Retrotransposon-based genetic variation of Poa annua populations from contrasting climate conditions

http://orcid.org/0000-0002-1851-3192 Androsiuk Piotr 1 piotr.androsiuk@uwm.edu.pl
Koc Justyna 1
http://orcid.org/0000-0001-6860-3666 Chwedorzewska Katarzyna Joanna 2
Górecki Ryszard 1
http://orcid.org/0000-0002-9001-1285 Giełwanowska Irena 1
1 Department of Plant Physiology, Genetics and Biotechnology, University of Warmia and Mazury in Olsztyn , Olsztyn , Poland
2 Department of Agronomy, Warsaw University of Life Sciences-SGGW , Warszawa , Poland
Escudero Marcial
Electronic publication date: 2019 May 15
Publication date: 2019
Volume: 7
Electronic Location ID: e6888
Received 2018 Oct 16; Accepted 2019 Apr 2
Copyright: © 2019 Androsiuk et al.
Copyright year: 2019
Copyright holder: Androsiuk et al.
License: This is an open access article distributed under the terms of the Creative Commons Attribution License, which permits unrestricted use, distribution, reproduction and adaptation in any medium and for any purpose provided that it is properly attributed. For attribution, the original author(s), title, publication source (PeerJ) and either DOI or URL of the article must be cited.
License URL: https://creativecommons.org/licenses/by/4.0/

Keywords: Annual bluegrass, Adaptation, Genetic diversity, Genetic structure, iPBS

Funding: The authors received no funding for this work.

==============================
Background

Poa annua L. is an example of a plant characterized by abundant, worldwide distribution from polar to equatorial regions. Due to its high plasticity and extraordinary expansiveness, P. annua is considered an invasive species capable of occupying and surviving in a wide range of habitats including pioneer zones, areas intensively transformed by human activities, remote subarctic meadows and even the Antarctic Peninsula region.

Methods

In the present study, we evaluated the utility of inter-primer binding site (iPBS) markers for assessing the genetic variation of P. annua populations representing contrasting environments from the worldwide range of this species. The electrophoretic patterns of polymerase chain reaction products obtained for each individual were used to estimate the genetic diversity and differentiation between populations.

Results

iPBS genotyping revealed a pattern of genetic variation differentiating the six studied P. annua populations characterized by their different climatic conditions. According to the analysis of molecular variance, the greatest genetic variation was recorded among populations, whereas 41.75% was observed between individuals within populations. The results of principal coordinates analysis (PCoA) and model-based clustering analysis showed a clear subdivision of analyzed populations. According to PCoA, populations from Siberia and the Kola Peninsula were the most different from each other and showed the lowest genetic variability. The application of STRUCTURE software confirmed the unique character of the population from the Kola Peninsula.

Discussion

The lowest variability of the Siberia population suggested that it was subjected to genetic drift. However, although demographic expansion was indicated by negative values of Fu’s FS statistic and analysis of mismatch distribution, it was not followed by significant traces of a bottleneck or a founder effect. For the Antarctic population, the observed level of genetic variation was surprisingly high, despite the observed significant traces of bottleneck/founder effect following demographic expansion, and was similar to that observed in populations from Poland and the Balkans. For the Antarctic population, the multiple introduction events from different sources are considered to be responsible for such an observation. Moreover, the results of STRUCTURE and PCoA showed that the P. annua from Antarctica has the highest genetic similarity to populations from Europe.

Conclusions

The observed polymorphism should be considered as a consequence of the joint influence of external abiotic stress and the selection process. Environmental changes, due to their ability to induce transposon activation, lead to the acceleration of evolutionary processes through the production of genetic variability.

Introduction

Vascular plants developed complex mechanisms to respond and adapt to recurring biotic and abiotic stresses (Bruce et al., 2007). There are a number of possible molecular mechanisms which may lead to genetically-determined phenotypic variability (Ingram & Bartels, 1996; Kreps et al., 2002; Tardif et al., 2007). In response to environmental stress, several types of morphological variants may arise, from which only the beneficial ones can be passed on to the next generation through natural selection (Piacentini et al., 2014). One of the mechanisms responsible for these evolutionary processes is transposon activation in response to severe environmental change which can disorganize the mechanism of transposon silencing (Kalendar et al., 2000; Piacentini et al., 2014). As a result, the explosion of transposon activity may be observed followed by the creation of genetic variability and associated macro-evolutionary processes (Rebollo et al., 2010; Schrader et al., 2014).

Annual bluegrass (Poa annua L.) representing the large Poaceae family consisting of 780 genera and around 12,000 species (Christenhusz & Byng, 2016) is an example of a plant species characterized by abundant, worldwide distribution (Fenner, 1985). It is spread from polar to equatorial regions and can be found in both natural habitats and cultivated ecosystems, where it is considered a weed. Due to its high plasticity and extraordinary expansiveness, P. annua is considered a colonizing species capable of occupying and surviving in wide range of habitats, including pioneer zones (Frenot, Gloaguen & Trehen, 1997), areas intensively transformed by human action, such as roadsides, pastures, gardens (Ellis, Lee & Calder, 1970); remote subarctic meadows (Heide, 2001) and the Antarctic Peninsula region (Chwedorzewska et al., 2015). According to the results of morphological analyses, hybridization trials, karyotype analyses and phylogenetic tests (Mao & Huff, 2012; Nannfeldt, 1937; Tutin, 1957; Soreng, Bull & Gillespie, 2010) P. annua is considered an allotetraploid hybrid (the most common chromosome count 2n = 28; Warwick, 1979) originating from two diploid parents, P. supina Schrad. and P. infirma Kunth (Mao & Huff, 2012; Tutin, 1952). Poa annua is preferentially an autogamous species with occasionally observed outbreeding, depending on environmental conditions (Ellis, 1973). Apomixes were also recorded for P. annua (Johnson et al., 1993). Reproductive biology and ecological studies have revealed P. annua is an extremely variable species which can grow as an annual (Grime, 1979), biennial (Tutin, 1957) or perennial plant (Lush, 1989; Till-Bottraud, Wu & Harding, 1990). Annual bluegrass grows and reproduces rapidly, primarily via seeds yielding up to 20,000 seeds in one season which may retain their viability for several years and forms a significant element of the soil seed bank (Hutchinson & Seymour, 1982).

Due to its world-wide distribution, polyploidy and heterozygous origin, P. annua is characterized by high variation in morphological characteristics and growth habits (Rudak et al., 2019; Galera, Chwedorzewska & Wódkiewicz, 2015). P. annua is able to develop adaptations depending on local climatic, edaphic and biotic conditions (Standifer & Wilson, 1988; Till-Bottraud, Wu & Harding, 1990; Rudak et al., 2019), which makes the species one of the most successful invasive species in the world (Randall, 2012). However, there is still a lack of sufficient knowledge of its genetics. To date, there have been few attempts to investigate the genetic variation of P. annua: e.g., Darmency et al. (1992) and Frenot et al. (1999) analyzed the inheritance of isozyme patterns while; Chen et al. (2003) and Li et al. (2004) investigated the traffic pollution impact on the isozyme polymorphism. Additionally, Mengistu, Mualler-Warrant & Barker (2000), Sweeney & Danneberger (1995) applied random amplified polymorphic DNA markers to study the relationship between wild P. annua collections and Carson, White & Smith (2007) used inter-simple sequence repeats (ISSR) markers to distinguish selected P. annua genotypes. Most recently, Chwedorzewska (2008), Chwedorzewska & Bednarek (2012) and Wódkiewicz et al. (2018) applied amplified fragment length polymorphism (AFLP) markers to study genetic polymorphism and the history of the species in the maritime Antarctic. However, there were no genetic studies on P. annua across the wide geographic range of the species.

The specific nature of the transposons (ubiquitous distribution, high copy number, widespread chromosomal dispersion, unique sequence features) enables the development of a number of multiplex DNA-based marker systems (Kalendar et al., 1999; Shedlock & Okada, 2000; Schulman, Flavell & Ellis, 2004), which are suitable for investigating genetic variability. The inter-primer binding site (iPBS) method is based on the virtually universal presence of a tRNA complement as a reverse transcriptase primer binding site (PBS) in long terminal repeat retrotransposons (Kalendar et al., 2010). This polymerase chain reaction (PCR)-based technique has been introduced as a powerful DNA fingerprinting technology which, contrary to previously developed transposon-based markers, can be applied without the need for prior sequence knowledge (Kalendar et al., 2010). The iPBS method, due to the length of applied primers and the high stringency achieved by the annealing temperature, appeared as a source of highly reproducible DNA markers with many potential applications, for example, clone identification (Baránek et al., 2012), genetic variation analyses (Guo et al., 2014; Fang-Yong & Ji-Hong, 2014; Koc et al., 2018) and phylogenetic studies (Özer, Bayraktar & Baloch, 2016). Moreover, these applications proved the iPBS method utility in assessing the scale of genome rearrangements in response to abiotic stress under local environmental conditions (Androsiuk et al., 2015).

In the present study, we evaluated the utility of iPBS markers for assessing the genetic variation of six P. annua populations collected from different environmental conditions: from the temperate climate of Central Europe, through the warm but seasonally dry southern part of the Old Continent, up to the harsh climate of western Siberia, northern Europe and maritime Antarctic. Furthermore, we investigated whether the selected method could be useful in establishing the link between assessed genetic variability of studied populations and their geographic origin and associated climate conditions.

Materials and Methods

Material

In order to realize the objectives of the study, six P. annua populations originating from different climatic conditions (Table 1; Fig. 1; Kottek et al., 2006) were selected. Populations from Albania (AL) and Macedonia (MA) represent stands characterized by hot summers, mild winters and precipitation occurring mostly in the cooler half of the year. Consequently they suffer from prolonged drought periods during the summer. The population from Poland (PO) is a stand with a climate characterized by the occurrence of four seasons. The seasons are easily recognizable and determined by the course of temperature (warm, humid spring; warm, usually dry summer; cool, humid autumn and winter, often with snow) with precipitation which occurs at different times of the year. Finally, populations from Siberia (SI) and the Kola Peninsula (KO) represent stands with very cold winters, warm but short summers, and varying precipitation throughout the whole year. However, SI and KO climates differ in some aspects but they can be regarded as examples of sites which can be defined by a harsh winter period. The latter population from Antarctica (King George Island, South Shetlands; AS), represent a polar habitat which, in addition to the characteristic severe winter periods, can also be defined by low-summer temperatures and high seasonal light regime (Frenot et al., 2005). However, the relatively short history of this species on King George Island (Olech, 1996; Olech & Chwedorzewska, 2011) and not entirely clear origin of this population (Chwedorzewska, 2008; Wódkiewicz et al., 2018) should be considered during the discussion of the results.

Figure 1 The geographic location of the studied sampling sites of Poa annua on a contour map of (A) Eurasia and (B) Antarctic.

Table 1 The origin of Poa annua populations used in the study and their population genetic characteristics.

Population	Sampling site	NB	P (%)	I	HE	
AS	King George Island, maritime Antarctica	140	14.97	0.091	0.063	
SI	Mukhrino, Western Siberia	137	6.12	0.032	0.022	
PO	Olsztyn, Poland	139	19.73	0.118	0.081	
MA	Ohrid, Macedonia	142	15.65	0.088	0.060	
AL	Saranda, Albania	144	19.05	0.108	0.073	
KO	Apatity, Kola Peninsula	132	12.93	0.081	0.057	
Mean over loci and populations		139.0	14.74	0.086	0.059	

DNA extraction and iPBS genotyping

The molecular analyses were performed using 143 P. annua individuals representing six populations (Table 1) ranging from 19 to 33 individuals per population. Tillers were collected as DNA material, cleaned, dried in silica gel and stored at –20 °C. Genomic DNA from each individual was extracted following the CTAB procedure with modifications by Murray & Thompson (1980) and Polok (2007). The quality of DNA was verified on 1% agarose gel (1 × TBE buffer with 0.5 μg/ml ethidium bromide), while the purity of DNA samples was assessed using NanoDrop (ND-1000 UV/Vis).

A total of 20 iPBS primers were tested according to the procedure described by Kalendar et al. (2010). Eight of them which gave polymorphic, clearly identifiable and repeatable bands were selected for further analyses (Table 2). The reproducibility of the band profiles for iPBS primers was tested by comparison of electrophoretic profiles for randomly selected P. annua samples. In this experiment, two replications of data were generated and compared. Gels were then checked to identify iPBS amplicons (bands) in only one or both replicates. Amplification was performed as described by Androsiuk et al. (2015). The PCR products were analyzed by electrophoresis in 1.5% agarose gel with 1 × TBE buffer at 100 V for 2 h and visualized by staining with 0.5 μg/ml ethidium bromide.

Table 2 iPBS primers applied in the study and their specification.

Primer	Sequence	Tm (°C)	Number of amplified bands	Number of polymorphic bands	
2085	5′-ATGCCGATACCA-3′	50	15	5	
2224	5′-ATCCTGGCAATGGAACCA-3′	52	19	7	
2229	5′-CGACCTGTTCTGATACCA-3′	56	25	11	
2231	5′-ACTTGGATGCTGATACCA-3′	52	14	5	
2238	5′-ACCTAGCTCATGATGCCA-3′	56	17	9	
2249	5′-AACCGACCTCTGATACCA-3′	58	23	13	
2253	5′-TCGAGGCTCTAGATACCA-3′	50	17	5	
2378	5′-GGTCCTCATCCA-3′	53	17	9	
Total	147	64	

Genetic diversity analyses using iPBS data

All bands that could be reliably read across all individuals were scored as either present (1) or absent (0) across genotypes and treated as single dominant loci. Based on the binary matrix obtained (Table S1), the following genetic parameters were estimated using GenAlEx 6.5 (Peakall & Smouse, 2006, 2012): total number of bands per population (NB), percentage of polymorphic bands (P), Shannon’s information index (I) and expected heterozygosity (HE).

Two methods were used to investigate the genetic structure of the samples. The first approach was the Bayesian model-based clustering method implemented in STRUCTURE ver. 2.3.4. (Pritchard, Stephens & Donnelly, 2000). The model assigns individual multilocus genotypes probabilistically to a user-defined number of clusters (K), achieving linkage equilibrium within clusters (Pritchard, Stephens & Donnelly, 2000). We conducted 10 replicate runs for each K, ranging from 1 to 10 (Fig. 2A). Each run consisted of a burn-in of 500,000 iterations, followed by data collection of over 2,000,000 iterations. The analysis using admixture model was conducted without any prior information on the original population. To determine the optimal number of clusters, an ad hoc statistic ΔK was used (Evanno, Regnaut & Goudet, 2005). The ΔK was evaluated in Structure Harvester ver. 0.6.94 (Earl & Vonholdt, 2012). The second method was a principal coordinates analysis (PCoA), based on the matrix of Euclidean distances between individuals from all analyzed populations, performed in PAST software (Hammer, Harper & Ryan, 2001).

Figure 2 The uppermost hierarchical level of genetic structure of studied Poa annua populations using STRUCTURE (Pritchard, Stephens & Donnelly, 2000).

(A) The values of the second-order rate of change of L(K), ΔK, of data between successive K values. (B) The population structure bar plots generated at K = 2.

Analysis of molecular variance (AMOVA) was performed with Arlequin 3.5. For this analysis, the iPBS data was treated as haplotypic, comprising of a combination of alleles at one or several loci (Excoffier, Laval & Schneider, 2005). The significance of the fixation indices was tested using a non-parametric permutation approach, the method implemented in Arlequin 3.5 (Excoffier, Smouse & Quattro, 1992; Excoffier, Laval & Schneider, 2005). Moreover, Tajima’s D, Fu’s FS neutrality test, and the mismatch distribution and demographic processes affecting populations were estimated using the same software. Bottleneck ver. 1.2.02 (Cornuet & Luikart, 1996) software was used to investigate recent effective population size reductions based on allele data frequencies (Cornuet & Luikart, 1996; Piry, Luikart & Cornuet, 1999) for each population. In populations that have experienced a recent reduction in their effective population size, the HE becomes larger than the heterozygosity expected at mutation-drift equilibrium. In order to study such effect using dominant markers, the infinite allele model (IAM) was used to test the mutation-drift vs bottleneck hypothesis (Tero et al., 2003). The significance of potential bottleneck was estimated using a sign test, standardized differences test and one-tailed Wilcoxon sign rank test for heterozygosity excess.

Results

Genetic diversity

Genetic analysis of all P. annua samples using eight iPBS primers identified 147 bands. The highest number of bands (25) was found in the iPBS2229 primer, whereas the lowest number (14) was scored for iPBS2231. The average number of bands per primer was 18.37. Out of all identified loci, 64 (43.5%) were polymorphic (Table 2). Detailed analysis revealed two private bands in the KO population: the first of them was revealed by iPBS2249 in 12 out of 33 individuals, and the second by iPBS2229 which was observed in all 33 individuals.

The iPBS markers revealed both the presence of genetic polymorphism between individuals within a population and genetic variation between populations. The number of iPBS bands ranged from 137 in the SI population, to 144 for AL population. The highest rate of polymorphic bands was scored for individuals from PO (19.73%) and AL (19.05%), whereas the lowest number of polymorphic bands was observed for the SI population (6.12%) (Table 1). The highest values of both Shannon’s information index and expected heterozygosity were observed for the PO population, whereas the lowest was observed for the SI population (Table 1).

Population genetic structure and differentiation

Bayesian clustering revealed that ΔK, the second-order rate of change of the likelihood function with respect to K, has a maximum at K = 2 (Fig. 2). Consequently, K = 2 was chosen as the optimal number of clusters of the uppermost hierarchical level of population structure. The first cluster consists of only one population from the Kola Peninsula, whereas the remaining five populations were gathered in the second cluster (Fig. 2B).

Principal coordinates analysis indicated that 50.69% of the variation was explained by the first three components (32.08%, 10.01% and 8.60%, respectively). Figure 3 illustrates the projection of the analyzed populations on the first two axes. The grouping revealed by PCoA showed that the most distinct character is represented by the KO population, which departed significantly along the first coordinate. The remaining five populations formed a heterogeneous group with partially overlapping clouds of individuals in which populations from MA, AN and PO were similar, but the population from AL formed a separate cluster (Fig. 3). Moreover, the individuals representing KO and SI populations formed the smallest and the densest clouds of individuals. The AMOVA results revealed that most of the described genetic variation occurred between populations (58.25%), whereas the variation among individuals within populations accounted for the remaining 41.75% (Table 3).

Figure 3 Plot of Coordinate 1 vs Coordinate 2 obtained by principal coordinates analysis (PCoA) based on Euclidean distances between all individuals from six Poa annua populations.

AS King George Island (filled circle), SI Mukhrino (inverted open triangle), PO Olsztyn (open square), MA Ohrid (open diamond), AL Saranda (times), KO Apatity (open triangle).

Table 3 Partitioning of diversity found in Poa annua populations using AMOVA (FST = 0.582).

Source of variation	d.f.	Sum of squares	Variance components	Percentage of variation	
Among populations	5	583.376	4.788	58.25	
Within populations	137	470.204	3.432	41.75	
Total	142	1,053.580	8.220		
Note:

Significance tests (1,023 permutations); p < 0.001.

Neutrality tests and demography

For the neutrality and demographic tests, Tajima’s D did not show any deviation from 0, while Fu’s FS was negative and significant for all populations (Table 4). In the mismatch distribution test for demographic/spatial expansion, there were no significant SSD values and all samples had a very low raggedness index (Table 5). The three heterozygosity excess tests (Sign test, Standardized differences test and Wilcoxon sign-rank test) produced significant p-values based on the IAM model, except for the SI population for which a lack of significant traces of bottleneck effect was observed (Table 6).

Table 4 Tajima’s D test and Fu’s FS neutrality tests for the analyzed populations.

Test	Description	Population	Statistics	
AS	SI	PO	MA	AL	KO	Mean	SD	
Tajima’s D test	S	22	9	29	23	28	19	21.667	7.257	
Pi	8.433	2.525	7.953	7.609	8.795	6.697	7.002	2.310	
Tajima’s D	1.317	0.157	0.091	0.869	0.386	1.465	0.714	0.593	
Tajima’s D p-value	0.918	0.625	0.615	0.845	0.704	0.948	0.776	0.147	
Fu’s FS test	Theta_pi	8.433	2.525	7.953	7.609	8.795	6.697	7.002	2.310	
Exp. no. of alleles	10.303	6.424	11.446	11.226	10.471	12.346	10.369	2.068	
FS	−7.406	−6.573	−16.159	−16.676	−12.401	−24.621	−13.973	6.721	
FS p-value	0.005	0.001	0.000	0.000	0.000	0.000	0.001	0.002	

Table 5 Mismatch analysis.

Model	Statistic	Populations	Mean	SD	
AS	SI	PO	MA	AL	KO	
Demographic expansion	SSD	0.021	0.002	0.003	0.002	0.008	0.002	0.006	0.008	
Model (SSD) p-value	0.290	0.730	0.590	0.860	0.460	0.670	0.600	0.203	
Raggedness index	0.018	0.042	0.014	0.009	0.014	0.012	0.018	0.012	
Raggedness p-value	0.790	0.620	0.590	0.880	0.660	0.600	0.690	0.118	
Spatial expansion	SSD	0.032	0.002	0.003	0.002	0.008	0.002	0.008	0.012	
Model (SSD) p-value	0.050	0.800	0.530	0.710	0.360	0.660	0.518	0.276	
Raggedness index	0.018	0.042	0.014	0.009	0.014	0.012	0.018	0.012	
Raggedness p-value	0.710	0.500	0.410	0.830	0.500	0.610	0.593	0.155	

Table 6 Testing the bottleneck vs mutation drift equilibrium hypotheses for all analyzed populations (IAM mutation model).

Population	Sign test	Standardized test	Wilcoxon test	
AS	HEEx = 10.02
HDe = 1
HEx = 20	T2 = 4.630
p = 0.0000	One tail for heterozygosity deficiency: 0.99999	
One tail for heterozygosity excess: 0.00001	
Two tails for heterozygosity excess and deficiency: 0.00003	
SI	HEEx = 3.66
HDe = 2
HEx = 6	T2 = 1.476
p = 0.06998	One tail for heterozygosity deficiency: 0.90234	
One tail for heterozygosity excess: 0.12500	
Two tails for heterozygosity excess and deficiency: 0.25000	
PO	HEEx = 12.68
HDe = 2
HEx = 26	T2 = 5.344
p = 0.0000	One tail for heterozygosity deficiency: 1.00000	
One tail for heterozygosity excess: 0.00000	
Two tails for heterozygosity excess and deficiency: 0.00000	
MA	HEEx = 9.91
HDe = 3
HEx = 19	T2 = 3.836
p = 0.00006	One tail for heterozygosity deficiency: 0.99996	
One tail for heterozygosity excess: 0.00004	
Two tails for heterozygosity excess and deficiency: 0.00009	
AL	HEEx = 12.87
HDe = 3
HEx = 24	T2 = 4.297
p = 0.00001	One tail for heterozygosity deficiency: 1.00000	
One tail for heterozygosity excess: 0.00000	
Two tails for heterozygosity excess and deficiency: 0.00000	
KO	HEEx = 7.82
HDe = 0
HEx = 18	T2 = 5.483
p = 0.00000	One tail for heterozygosity deficiency: 1.00000	
One tail for heterozygosity excess: 0.00000	
Two tails for heterozygosity excess and deficiency: 0.00000	
Note:

HEEx, expected heterozygosity excess; HDe, heterozygosity deficiency; HEx, heterozygosity excess.

Discussion

Poa annua is an excellent example of an expansive species able to grow in very wide range of habitats. This inconspicuous plant possesses a number of features (e.g., small size, short life cycle, rapid germination, tolerance to frost, grazing and trampling) which allow it to expand its distribution and to adapt to broad array of climatic conditions (Law, 1981; Grime, Hodgson & Hunt, 1986; Frenot, Gloaguen & Trehen, 1997; Holm et al., 1997; Mitich, 1998; Vargas & Turgeon, 2004). Consequently, P. annua, which can be found from the cold polar regions to the hot deserts, is known to exhibit high morphological variation (Galera, Chwedorzewska & Wódkiewicz, 2015; Williams et al., 2018). Although morphological variability was reported (Darmency & Gasquez, 1983; Galera, Chwedorzewska & Wódkiewicz, 2015; Williams et al., 2018), available literature describing the genetic variation of P. annua is scarce. Moreover, these genetic studies generally use a low number of analyzed populations or outdated techniques. On the one hand, there are data based on isozyme and allozyme polymorphism which point to weak genetic polymorphism with a predominant homozygosity in study populations (Darmency et al., 1992; Frenot et al., 1999). On the other hand, the abundance of polymorphic loci is reported using AFLP (up to 60%; Chwedorzewska, 2008) or ISSR markers (77.3%; Carson, White & Smith, 2007). However, the conclusions drawn from the direct comparison of the results mentioned above should be treated with caution since the polymorphism revealed by isozymes is limited due to the small number of loci (with rather low allelic diversity) that could be obtained per individual, moreover, isozymes play certain metabolic functions and therefore weak selection pressure is expected. To the contrary, DNA markers (especially AFLP) have a tendency to generate a high number of polymorphic fragments which are generally selectively neutral (Vos et al., 1995).

In our studies, iPBS markers revealed 43.5% of polymorphic bands for all analyzed populations. However, when each population is considered individually, the average polymorphism reached only 14.74% (Table 1). Moreover, the observed polymorphism appeared to be unevenly distributed among analyzed populations—the value of this parameter ranged from 19.73% in PO and 19.05% in AL, to 6.12% in SI. The previous applications of iPBS markers showed that the level of polymorphism revealed by that molecular tool may vary from 97.4% (for Myrica rubra; Fang-Yong & Ji-Hong, 2014) to 4.88% (polymorphism found among clones of the apricot cultivars; Baránek et al., 2012).

The AMOVA revealed that almost two-thirds of genetic variation was recorded among populations (Table 3). Surprisingly, previous studies on P. annua using different markers reported that most of genetic variance was detected within analyzed populations (Chwedorzewska, 2008). For example, comparative studies of samples from King George Island (South Shetland Islands, western Antarctica), Argentina (Ushuaia) and Poland (Dziekanów Leśny) based on AFLP markers revealed that 30% of the detected variability is distributed among populations (Chwedorzewska, 2008). Similar results were obtained by Frenot et al. (1999) in their studies on genetic variation between populations from France and sub-Antarctic: Crozet and Kerguelen Islands based on isoenzyme polymorphism. They also found that most of the variation is located within populations, on average 65%. Nevertheless, a detailed study of the analyses mentioned above shows that for half of the applied enzyme systems, genetic variation found within the population did not exceed 56% or 50% of the total genetic variation. Additionally, there are results obtained using metAFLP (molecular markers which allow tracing the epigenetic variation in the genome), which also showed that ca. 50% of genetic variation is located between P. annua populations (Chwedorzewska & Bednarek, 2012). Furthermore, AFLP markers applied to the same set of analyzed populations revealed that only 26.7% of genetic variation is distributed between them (Chwedorzewska & Bednarek, 2012). The higher epigenetic variation in response to external, abiotic stress factors confirms the crucial role of epigenetic components of the genome in long-term survival under unfavorable conditions like low temperature (Stewards et al., 2002), water deficit (Labra et al., 2002) and osmotic stress (Tan, 2010). Analogically to the character of epigenetic changes in the genome, the polymorphism revealed by iPBS markers (based on transposable elements and their mobile character) is also shaped in response to various abiotic stresses (Capy et al., 2000; Schrader et al., 2014; Makarevitch et al., 2015). Therefore, a congruent pattern of genetic variation partition obtained in the present study and by metAFLP may be explained by the similar nature of both molecular techniques which appeared suitable for tracing the changes in the genome which arise due to the adaptation process (Chwedorzewska & Bednarek, 2012; Androsiuk et al., 2015).

The results of PCoA and model-based clustering analysis showed that there was clear genetic subdivision within the six populations of P. annua. According to PCoA, two populations: KO and SI were different from each other and were characterized by the lowest genetic variability among the studied populations. STRUCTURE software confirmed the population subdivision identifying the individual character of P. annua from Kola Peninsula, which forms a separate cluster. Moreover, populations KO and SI are characterized by the lowest values of all genetic characteristics among the whole collection of P. annua. Especially the lowest variability of SI population suggested that it was subjected to genetic drift. Although demographic expansion was indicated by negative Fu’s FS values and an analysis of mismatch distribution, it was not followed by significant traces of a bottleneck or founder effect. Therefore, putative population selection processes should be considered to explain that observation. Long-term selection pressure from a consistent factor or group of factors may result in the narrowing of genetic variation favoring specific genotypes. This mechanism of adaptation was observed by Darmency & Gasquez (1983) and Mengistu, Mualler-Warrant & Barker (2000) who observed narrowed diversity in P. annua populations which experienced long herbicide selection pressure. In the area of Western Siberia, the most important selection forces limiting plant survival and expansion are associated with climatic conditions typical of this zone. As a consequence, the development of freezing tolerance becomes a key feature which enables the survival of P. annua individuals of some ecotypes even at temperatures as low as −31.6 °C of median lethal temperature (Dionne et al., 2001). Under stress conditions, P. annua shows slower development, greater competitive ability, delayed reproduction, greater biomass per individual, continued reproduction and domination of perennial ecotypes in the population (Law, Bradshaw & Putwain, 1977; Galera, Chwedorzewska & Wódkiewicz, 2015). Moreover, it can produce thousands of seeds per individual which survive in the soil under unfavorable conditions and may retain the capability of germination for several years, forming a soil seed bank (Law, 1981; Hutchinson & Seymour, 1982). This survival strategy was found for P. annua in both sub-Antarctic and Antarctic conditions (Walton & Smith, 1973; Wódkiewicz et al., 2013, 2014) and an analogous scheme should therefore also be expected for Siberia.

The unique character of plants found in polar regions is reflected in many morphophysiological traits and can be traced in all stages of their growth, including the very early stages of ontogenesis. Our previous studies of the content and composition of soluble carbohydrates during seed development in P. annua from the Antarctic (King George Island) and Poland (Olsztyn) revealed both qualitative and quantitative differences (Kellmann-Sopyła & Giełwanowska, 2015). This variation can be interpreted as one of the possible mechanisms of freezing tolerance as well as adaptation to other environmental stresses, for example, water stress, a specific light regime and elevated ultraviolet-B radiation levels which are characteristic of polar regions. The extremely high concentration of fructose in mature caryopses of P. annua, originating from the Antarctic, may reveal a metabolic feature indicating the readiness of caryopses to immediate germination in a short period of favorable growing conditions (Kellmann-Sopyła & Giełwanowska, 2015). For the Antarctic population, the observed level of genetic variation was surprisingly high, despite the observed bottleneck/founder effect followed by demographic expansion, and was similar to that observed in populations from Poland and the Balkans. These results are in accordance with previous observations concerning the Antarctic population of P. annua presented in Chwedorzewska (2008) and Chwedorzewska & Bednarek (2012) publications. The multiple introduction events from different sources are considered to be responsible for such an observation (Lityńska-Zając et al., 2012; Chwedorzewska et al., 2013, 2015).

Poa annua from the populations representing the central part of the species range in Europe (populations MA, AL and PO) appeared to share a considerably high degree of similarity (as shown by PCoA results). However, although the AL population departed slightly from PO and AM populations along the first coordinate, STRUCTURE failed to detect any genetic structure among them. The observed data structuring is characteristic of the populations from the central part of the species distribution, where gene flow, varying in rate and direction, may counteract the effects of putative demographic and selection processes.

According to the obtained results, P. annua from the KO population deserves special attention. As with population SI, it is characterized by a lower-than-average level of genetic variation, expressed in the number of polymorphic loci and expected heterozygosity (Table 1) although it is the only population studied for which private alleles were identified. The occurrence of these population-specific bands undoubtedly contributes to the unique character of this population, which is clearly visible on a PCoA diagram, as well as in the results of Bayesian clustering. These specific alleles may be an example of unique mutations which arose as a consequence of transposon activation and persist only in these particular populations, perhaps due to their beneficial character. However, their presence in the KO population may be also interpreted as an example of selection favoring rare genotypes which can also be found in other P. annua individuals from neighboring populations which were not sampled during the realization of this project. This indicates that the presence of such unique alleles in the KO population may suggest its different historical dispersal, that is, origination from different glacial refugia. Although the Iberian, Italian and Balkan peninsulas are believed to be most suitable environments to harbor a large fraction of the interspecific biodiversity of the temperate biota during the glacial period, there are reports based on fossil pollen data and macrofossil remains which indicate that such refugee areas may also be placed at the southern edge of cold and dry steppe-tundra areas in eastern, central and southwestern Europe (Bennett, Tzedakis & Willis, 1991; Willis, Rudner & Sümegi, 2000; Carcaillet & Vernet, 2001). However, in order to verify this hypothesis, enhanced sampling of P. annua populations from eastern and north-eastern Europe is needed. The very limited data on the genetic variability of P. annua also makes it difficult to resolve this problem.

Conclusions

Inter-primer binding site markers revealed an interesting pattern of genetic variation among studied populations of P. annua representing locations diversified in terms of climatic conditions. The observed polymorphism should be considered to be a consequence of the combined influence of external abiotic stress and the selection process. Environmental changes, due to their ability to induce transposon activation, lead to the acceleration of evolutionary processes through the production of genetic variability. To the contrary, selection pressure from external factors (e.g., environmental conditions) or the purification of a selection against deleterious alleles, hampers the evolution rate. As a consequence, balanced selection selects the optimal compromise among these constraints. The wide geographic range of P. annua is a result of the outstanding phenotypic and physiological plasticity of this species, which leads to its high ecotypic differentiation in response to various limiting environmental factors in each habitat. However, the answer to the question of whether the ecological breadth of the species is a consequence of such a specialist strategy or is possible due to the existence of plastic “multi-purpose” genotypes (Sultan, 1995) seems to be an interesting subject for further studies.

Supplemental Information

Supplemental Information 1 Binary matrix of amplification products revealed by eight iPBS primers for 143 individuals of Poa annua.

Click here for additional data file.

Supplemental Information 2 iPBS electrophoresis.

Click here for additional data file.

Additional Information and Declarations

Competing Interests

Author Contributions

Data Availability

The authors declare that they have no competing interests.

Piotr Androsiuk conceived and designed the experiments, performed the experiments, analyzed the data, prepared figures and/or tables, authored or reviewed drafts of the paper, approved the final draft.

Justyna Koc performed the experiments, approved the final draft.

Katarzyna Joanna Chwedorzewska prepared figures and/or tables, authored or reviewed drafts of the paper, approved the final draft.

Ryszard Górecki contributed reagents/materials/analysis tools, approved the final draft.

Irena Giełwanowska conceived and designed the experiments, contributed reagents/materials/analysis tools, approved the final draft.

The following information was supplied regarding data availability:

The raw genotyping data are available in Table S1 and at DOI 10.5281/zenodo.2640005 where the raw electrophoresis data were deposited.

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
