# Peer review of "Retrotransposon-based genetic variation of Poa annua populations from contrasting climate conditions"

_PeerJ, doi:10.7717/peerj.6888_

## Round 0.1 · original submission · Major Revisions

Dear Piotr,

The reviewers agree that your study could be valid for publication in PeerJ. However, the reviewers have many suggestions and several major concerns (see specially the comment from reviewer 1). The reviewer 3 considers that the discussion might be much improved.

I recommend major revision. The submission of a new improved version is welcome if you could satisfy reviewers' concerns.

Cheers,
Marcial.

Reviewer 1 ·

Basic reporting

In this work, it was used inter primer binding sites (iPBS) technique for study the genetic polymorphism within selected Poa annua populations in terms of climatic conditions characteristic for the sampling sites.

Experimental design

On the content of the manuscript and the quality of the analysis - there are no comments.

Validity of the findings

However, the authors did not provide any electrophoresis pictures, therefore, the manuscript cannot be considered for publication in the current form, since the primary data is not presented.

Additional comments

Authors need to add a few pictures of electrophoresis for several iPBS primers for all Poa annua samples to assess the quality of the primary data. Otherwise, the article should be rejected.
Based on the quality of these PCR-fingerprinting pictures, it will be possible to evaluate the quality of the subsequent analysis and the work as a whole.

Reviewer 2 ·

Basic reporting

The manuscript is considered with interesting information. Its writing is clear, and its possible publication is recommended. The information considered as the introduction, and discussion is appropriate. Materials and Methods are clear. The results are adequate, and the Tables and Figures are also appropriate. Regarding the bibliographic citations, it is suggested to attend the following observations because it is not referenced correctly:

a) there are bibliographic citations referred to throughout the text that is not in the list of references and vice versa.
b) there are citations throughout the text constituted by the three authors; however, some include the full last name of these three authors followed by the year, and in other cases only the first author is cited followed by "et al.". This latest form of citation is the one recommended in scientific literature.
c) some citations from the list of references have the year in which they were published in cursive letters and there is no reason for that.
d) check that book citations in the references list, include the pages consulted.

Experimental design

The objective of the research is clear and demonstrated with the experimental design developed. The results were analyzed in an appropriate way, which allowed the biological interpretation of them. However, the authors do not indicate in the Material and Methods sections if repetitions of the PCR reactions or the electrophoretic runs were performed to check their repeatability. It is recommended indicate this aspect.

Validity of the findings

The validation; In this case, the interpretation for the results is clear through the various analyzes carried out. The discussion was conducted correctly by comparing the results obtained in the present investigation with information from other researchers.
However, the conclusions included a bibliographical citation: the conclusions do not include citations.

Additional comments

Too many bibliographic citations were included, which makes the reading and fluency of the article is not adequate. Is it possible to suggest to the authors that they only consider those most relevant citations?

Reviewer 3 ·

Basic reporting

The basic reporting of the results in the manuscript is OK. I have a feeling that it can be improved. The language seem to be a bit of a challenge and it need to be improved to clarify many aspects of the MS.

Experimental design

The experimental design was set up very well. Enough coverage of the regions with good sample sized. It includes samples from both the native and introduced ranges which is good for a population genetic analysis and adaptation study.

Validity of the findings

The study has good results but the interpretation and validation can be improved. It doesn’t seem like the authors understand fully the kind of question they are trying to answer. So far the MS has no focal point. I feel like certain analysis and not necessary but this would be clearer if the authors can clearly state the questions and hypotheses they are testing here.

Additional comments

The manuscript entitled “Retrotransposon-based genetic variation of Poa annua populations from contrasting climate conditions” is a very interesting study which highlights both aspects of population genetics, ecology and adaptation. The authors used a good DNA marker for the investigation. The analyses were done adequately and the authors have a good understanding of the data. However, the MS lacks a focal point. Therefore, the results are hard to validate. Moreover, the DISCUSSION session needs a lot of improvement. The tables and figures also need to be improved. The English language seem to be a challenge and it needs to be improved to clarify most claims made by the authors. I feel like this is a very important study to illustrate aspects of adaptation during colonization and range expansion. But the authors didn’t discuss much about adaptation. There are a lot of grammatical errors which need to be corrected.

Annotated reviews are not available for download in order to protect the identity of reviewers who chose to remain anonymous.

---

## Round 0.2 · Minor Revisions

Dear Piotr,

The three reviewers conclude that the manuscript has been much improved. Please, consider the suggestions and submit a new improved version.

Sincerely,
Marcial.

Reviewer 1 ·

Basic reporting

All comments have been addressed

Experimental design

The study of electrophoresis pictures of PCR samples, I can say. Of course, the author basically received adequate PCR with these primers and DNA samples. However, not everything is perfect, due to which the analysis could have this data may contain errors.
The authors could improve the quality of separation of PCR bands with a long gel and image acquisition not with a digital camera, but with more sensitive equipment, or through a red filter and a long exposure.

Validity of the findings

no comments

Reviewer 2 ·

Basic reporting

My suggestions were taken care of correctly. It is only suggested to check the following: on line 260 the word " warm " is repeated.

Experimental design

No comments

Validity of the findings

No comments

Additional comments

It is only suggested to check the following: on line 260 the word " warm " is repeated.

Reviewer 3 ·

Basic reporting

The basic reporting of the results in the manuscript has been improved to my satisfaction, except for the Fig 2 which needs to be clarified. The ∆K figure is WRONG. The authors must submit the appropriate figure if this paper has to be published.

Experimental design

The experimental design was set up very well. Enough coverage of the regions with good sample sized. It includes samples from both the native and introduced ranges which is good for a population genetic analysis and adaptation study.

Validity of the findings

The study has good results and interpretation of the results has been made clear.

Additional comments

The manuscript entitled “Retrotransposon-based genetic variation of Poa annua populations from contrasting climate conditions” is a very interesting study which highlights both aspects of population genetics, ecology and adaptation. The authors used a good DNA marker for the investigation. The analyses were done adequately and the authors have a good understanding of the data. The improved MS now has a focal point. Moreover, the DISCUSSION section has been improvement adequately. The tables and figures legends also have been improved.

Annotated reviews are not available for download in order to protect the identity of reviewers who chose to remain anonymous.

---

## Round 0.3 · Minor Revisions

Dear Piotr,

Congratulations! Your study is almost now ready for publication in PeerJ.

Consider the minor suggestion by the reviewer and submit the final version.

Sincerely,

Marcial.

Reviewer 3 ·

Basic reporting

The basic reporting of the whole manuscript has been improved to my satisfaction and the paper is now clear and concise.

Experimental design

The experimental design was set up very well. Enough coverage of the regions with good sample size. It includes samples from both the native and introduced ranges which is good for a population genetic analysis and adaptation study.

Validity of the findings

The results are valid with the use of an up to date technique to answer the question.

Additional comments

The manuscript entitled “Retrotransposon-based genetic variation of Poa annua populations from contrasting climate conditions” has been improved adequately by addressing all the comments. I am satisfied with the way everything has been presented. Kindly address the very minor comments I have. Well done and all the best.

Annotated reviews are not available for download in order to protect the identity of reviewers who chose to remain anonymous.

---

## Round 0.4 · accepted · Accept

Dear Piotr,

Congratulations!

Cheers,

Marcial.

#